# A Prospective Analysis of the Burden of Multi-Drug-Resistant Pathogens in Acute Appendicitis and Their Implication for Clinical Management

**DOI:** 10.3390/antibiotics14040378

**Published:** 2025-04-04

**Authors:** Faruk Koca, Sara Fatima Faqar-Uz-Zaman, Claudia Reinheimer, Michael Hogardt, Volkhard A. J. Kempf, Paul Ziegler, Andreas A. Schnitzbauer, Armin Wiegering, Wolf Otto Bechstein, Patrizia Malkomes

**Affiliations:** 1Department for General, Visceral, Transplant and Thoracic Surgery, Hospital of the Goethe University Frankfurt, 60590 Frankfurt am Main, Germany; faruk.koca@unimedizin-ffm.de (F.K.); sarafatima189@gmail.com (S.F.F.-U.-Z.); armin.wiegering@unimedizin-ffm.de (A.W.); bechstein@med.uni-frankfurt.de (W.O.B.); 2Institute for Medical Microbiology and Infection Control, Hospital of the Goethe University Frankfurt, 60590 Frankfurt am Main, Germany; claudia.reinheimer@unimedizin-ffm.de (C.R.); michael.hogardt@unimedizin-ffm.de (M.H.); volkhard.kempf@unimedizin-ffm.de (V.A.J.K.); 3Senckenberg Institute of Pathology, Hospital of the Goethe University Frankfurt, 60590 Frankfurt am Main, Germany; paul.ziegler@unimedizin-ffm.de; 4Department of Surgery, Knappschaft Kliniken University Hospital Bochum, Ruhr-University Bochum, In der Schornau 23-25, 44892 Bochum, Germany; andreas.schnitzbauer@knappschaft-kliniken.de

**Keywords:** appendicitis, multi-drug-resistant pathogens, antibiotic resistance, intraoperative swab

## Abstract

**Background/Objectives**: Appendicitis caused by multi-drug-resistant pathogens is associated with significant postoperative morbidity. However, prospective data on the microbial spectrum and its clinical impact remain limited. **Methods**: Adults with acute appendicitis undergoing surgery between April 2022 and July 2023 were prospectively enrolled at a single university-affiliated institution. Bacterial cultures from appendiceal and rectal swabs were analyzed, and clinical outcomes were assessed. A telephone follow-up was conducted 30 days postoperatively. **Results**: A total of 105 patients were included. Multi-drug-resistant pathogens were identified in the appendiceal swabs of twenty-nine patients (27.6%), while six patients (5.7%) harbored multi-drug-resistant organisms (MDROs; according to the criteria of the CDC). Rectal swabs revealed MDROs in 11.4% of cases but showed a limited correlation with appendiceal samples, indicating that rectal colonization does not reliably predict the presence of MDROs in appendicitis. Patients with multi-drug-resistant infections had significantly higher postoperative complication rates (31% vs. 10.5%, *p* = 0.017), including more Clavien–Dindo grade 3 complications (17.2% vs. 2.6%, *p* = 0.007) and abdominal abscesses (10.3% vs. 1.3%, *p* = 0.03). These patients required more frequent postoperative antibiotic treatment (65.5% vs. 40.8%, *p* = 0.03) and therapy adjustments (37.9% vs. 15.8%, *p* = 0.02). Hospital stays were also prolonged in the multi-drug-resistant group (a median of 4 days and IQR of 5 days vs. a median of 3 days and IQR of 3 days; *p* = 0.03). **Conclusions**: Colonization with multi-drug-resistant pathogens in appendicitis is associated with worse clinical outcomes. The intraoperative microbiological analysis of appendiceal swabs in complicated cases may enable targeted antibiotic therapy, potentially shortening hospital stays, optimizing patient management and reducing healthcare costs.

## 1. Introduction

Acute appendicitis (AA) is a common intra-abdominal infection, primarily caused by the obstruction of the appendiceal lumen due to factors such as appendicoliths (the accumulation of feces in the lumen of the appendix), lymphoid hyperplasia, swelling, or, in rare cases, benign and malignant tumors. This obstruction leads to bacterial overgrowth, resulting in significant morbidity [1,2,3,4]. The role of multi-drug-resistant organisms (MDROs) in abdominal surgery and subsequent surgical site infections (SSIs) is of increasing concern, as they complicate treatment and postoperative recovery [5].

Laparoscopic appendectomy remains the gold standard for treating AA, offering reduced morbidity and faster recovery [1]. However, postoperative complications occur in 2% to 23% of cases, with surgical site infections (SSIs) being the most frequent and clinically significant [6]. Approximately 4.3% of patients require reoperation or invasive interventions due to infectious complications [7].

Antibiotics play a crucial role in both preventing and managing these complications, but current antimicrobial guidelines are based on outdated microbiological data. The increasing adoption of non-operative medical treatments for uncomplicated AA further underscores the need for updated insights into the microbial landscape.

While perioperative antibiotic prophylaxis is standard practice for preventing SSIs, the necessity and optimal duration of postoperative antibiotic therapy remain debated [8]. Reports suggest that about 40% of patients with uncomplicated AA and 48% with complicated AA receive postoperative antibiotics, yet the impact of this treatment on SSI rates is inconsistent, making the requirement for postoperative antibiotics unclear [9,10]. Furthermore, the global rise in MDROs complicates treatment. The prevalence of MDROs has been reported at 6.4% in uncomplicated and 11.2% in complicated AA [11,12].

Moreover, patients harboring MDROs in appendiceal swabs are at an increased risk of infectious complications, suggesting that the microbial spectrum may influence postoperative outcomes beyond the complexity of appendicitis alone [11].

Despite these findings, data characterizing the microbiological spectrum in AA remain limited and predominantly retrospective. A detailed prospective evaluation of multi-drug-resistant pathogens and their antibiotic susceptibility in adult patients with AA is lacking in the literature. While a correlation between SSIs and rectal colonization has been observed after abdominal surgery [5], the relationship between appendicitis and rectal colonization remains unclear. Furthermore, it is not well established whether multi-drug-resistant pathogens in AA influence the clinical course and postoperative outcomes.

Therefore, this prospective study aimed to characterize the pathogen landscape in AA with a focus on resistant strains and to evaluate their impact on clinical outcomes and postoperative recovery. By doing so, we sought to refine antibiotic management strategies and improve the treatment of AA in the context of rising antibiotic resistance.

## 2. Results

From April 2022 to July 2023, a total of 108 adult patients with suspected acute appendicitis underwent surgical treatment at the University Hospital Frankfurt, Germany. Based on pathologic findings, three patients were excluded from the study due to not meeting the histopathologic criteria for appendicitis. Of the 105 patients enrolled, 42 were female (40%). The median age was 34 years (IQR: 16 years).

### 2.1. Incidence of MDROs in Rectal and Appendiceal Swab Specimens

Twelve patients (11.4%) tested positive for MDROs in the rectal swab screening, while six patients (5.7%) had MDROs detected in appendiceal swabs. In total, fourteen patients (13.3%) were found to have MDROs, with only four cases showing the same pathogen in both swabs. Accordingly, in ten cases, either the rectal or appendiceal swab tested positive for MDROs (Table 1).

### 2.2. Microbiological Spectrum and Antibiotic Resistance in Appendiceal Swabs

Bacteria were isolated from 71 of 105 specimens sent for culture (67.6%), while 32.4% of the swabs remained sterile. A total of 15 bacterial species were identified. The most frequently cultured organisms were *Escherichia coli* (46.7%), *Bacteroides* spp. (30.4%), *Enterococcus* spp. (17.1%), *Streptococcus* spp. (5%), and *Pseudomonas aeruginosa* (7.6%) (Table 2). Fungi were isolated from the swabs of five patients (4.8%). In 20 patients (19%), more than one microorganism was identified. Antibiotic susceptibility testing (AST) revealed that 59 pathogens (56.1%) exhibited no antibiotic resistance (Table 3). The most common resistance was to ampicillin (n = 53, 50.5%). In 37 patients (35.2%), pathogens showed “susceptible increased exposure” to cefuroxime, as defined by the European Committee on Antimicrobial Susceptibility Testing (EUCAST), indicating a high probability of therapeutic success with adjusted dosing [13,14].

The study population was divided into two groups: the non-MDR group, consisting of 76 patients with no detectable microorganisms, no antibiotic resistance, or resistance to a single antibiotic, and the MDR group, consisting of 29 patients with bacteria resistant to at least two antibiotics. Among the MDR group, six patients met the CDC criteria for MDROs (Table 3).

### 2.3. Diagnostics and Clinical Findings

The clinical and intraoperative findings are presented in Table 4. There were no statistically significant differences in the clinical and laboratory findings between the MDR and non-MDR groups. Imaging via a CT or MRI scan was performed with a similar frequency in both groups. Surgery was performed more than 24 h after the symptom onset in 77 patients (73.3%), with a significantly higher proportion of delays in the MDR group (26 patients, 89.7%) compared to the non-MDR group (51 patients, 67.1%). Thus, the duration from the symptom onset to surgery was significantly longer in the MDR group compared to the non-MDR group (*p* = 0.023). Uncomplicated appendicitis was observed in 70 patients (66.7%), while 35 patients (33.3%) had complicated appendicitis, including 18 patients (17.1%) with perforation, eight (7.6%) with distal necrosis, and ten (9.6%) with an abscess. Primary laparotomy was required in two patients (1.9%), and a conversion to laparotomy occurred in three patients (2.9%). Although the MDR group exhibited a higher incidence of complicated appendicitis (13 patients, 44.8%) compared to the non-MDR group (22 patients, 28.9%), this difference did not reach statistical significance (*p* = 0.09, Table 4).

### 2.4. Postoperative Complications and Antibiotic Treatment

Postoperative complications were categorized according to the Clavien–Dindo classification, as summarized in Table 5 [15].

Postoperative complications occurred in seventeen patients (16.2%), including two patients with grade 3b complications and five patients with grade 3a complications according to the Clavien–Dindo classification. Both grade 3b complications required secondary abdominal wall closure under general anesthesia due to deep SSIs and fascial dehiscence. Five patients underwent interventional abscess drainage. Two patients experienced non-infectious postoperative complications: one developed an intra-abdominal hematoma, which was managed conservatively, while the other had postoperative bowel atony, which was treated with laxatives. No grade 4 or 5 complications were observed.

SSIs occurred in 15 patients (14.3%). The MDR group had a significantly higher incidence of Clavien–Dindo grade 3 complications compared to the non-MDR group (17.2% vs. 2.6%, *p* = 0.007). Furthermore, postoperative abdominal abscesses, classified as A3 SSIs according to the CDC classification system, were significantly more frequent in the MDR cohort in comparison to the non-MDR group (10.3% vs. 1.3%; *p* = 0.031, Table 6).

The median length of the hospital stay was 3 days (IQR: 3 days). In the MDR group, the mean length of the stay was significantly longer (median: 4 days; IQR: 5 days) compared to that of the non-MDR group (median: 3 days; IQR: 3 days; *p* = 0.028). All patients received single-dose antibiotic prophylaxis, with 95 patients (90.5%) receiving cefuroxime and metronidazole 30–60 min before skin incision. Postoperative antibiotic therapy was administered to 50 patients (47.6%) based on intraoperative findings, such as peritonitis, perforation, or an abscess. A significantly higher percentage of MDR patients required postoperative antibiotics in comparison to the non-MDR group (65.5% vs. 40.8%, *p* = 0.03). Antibiotic therapy was adjusted after receiving susceptibility results in 23 patients (21.9%), with a higher adjustment rate in the MDR group (n = 11, 37.9%) compared to the non-MDR group (n = 12, 15.8%; *p* = 0.02). Thirteen patients (12.4%) were switched to oral antibiotics, and thirteen patients required an escalation to intravenous therapy. Antibiotic escalation occurred significantly more often in the MDR group (27.6%) compared to the non-MDR group (6.6%; *p* = 0.007). Similarly, the use of reserve antibiotics was more frequent in the MDR group (13.8%) than in the non-MDR group (2.6%; *p* = 0.048, Table 6).

## 3. Discussion

AA affects all age groups, with laparoscopic appendectomy remaining the gold-standard treatment. However, non-operative management with antibiotics has gained traction, particularly for uncomplicated appendicitis. Despite this trend, approximately 30% of patients initially treated conservatively progress to complicated appendicitis, and 14% require secondary intervention due to an inadequate response to empirical antimicrobial therapy [16,17]. Additionally, up to 20% of patients undergoing appendectomy develop postoperative complications, predominantly infections [17]. Given the increasing shift towards non-operative management, a precise understanding of the microbial spectrum and the role of resistant pathogens is essential for optimizing both initial therapy and the management of intra-abdominal infections following appendicitis or appendectomy.

In our prospective study of 105 patients, bacteria were isolated from 67.6% of appendiceal swabs. The incidence of MDROs, as defined by the CDC, was 5.7% in appendiceal swabs and 11.4% in rectal swabs, with no significant correlation between the two. Overall, 27.6% of patients had AA associated with multi-drug-resistant (MDR) pathogens. Patients in the MDR group experienced significantly higher rates of postoperative complications, predominantly infections, including more severe SSIs and abdominal abscesses. Furthermore, these patients required more frequent antibiotic adjustments and had prolonged hospital stays. Importantly, only 11.4% of patients exhibited primary resistance to the standard perioperative prophylaxis with cefuroxime and metronidazole.

Most previous studies on appendicitis microbiology were retrospective and primarily focused on pediatric populations. Moreover, most studies analyzed peritoneal fluid rather than appendix tissue, which may have limited their relevance for guiding antibiotic therapy [18,19]. Our study, based on direct appendiceal swabs, identified pathogens in 67.6% of cases, a significantly higher detection rate compared to the 50% reported in peritoneal fluid-based studies. *E. coli* (46.7%) and *Bacteroides* spp. (30.4%) were the most frequently isolated pathogens, consistent with previous findings [11,19]. The MDRO prevalence varies geographically, with reported rates ranging from 4.4% in Europe to 28.0% in the Asia–Pacific region [20,21]. The incidence of MDROs in our study (5.7% in appendiceal swabs and 11.4% in rectal swabs) was in line with European data [11]. However, rectal swabs showed poor predictive value for an MDRO presence in appendiceal samples (sensitivity: 33.3%; specificity: 86.7%), suggesting that routine rectal screening is not a reliable tool for identifying patients at risk for antibiotic-resistant infections.

In our cohort, the presence of multi-drug resistant pathogens correlated with a symptom duration exceeding 24 h before surgery, supporting the hypothesis that delayed intervention may contribute to shifts in the microbial spectrum and increased disease severity. Additionally, patients with complicated appendicitis exhibited higher antibiotic resistance rates (60% vs. 35.7%, *p* = 0.02), suggesting a role of resistant bacteria in disease progression.

Postoperative complications occurred in 16.2% of patients, with SSIs being the most common (88.3%). Non-infectious complications were rare, with only two patients developing an intra-abdominal hematoma or postoperative bowel atony, both managed conservatively. Our analyses indicate that antibiotic resistance influences both the incidence and severity of SSIs. Patients in the MDR group had significantly higher rates of CDC A3 SSIs, requiring more frequent wound revisions and prolonged antibiotic therapy. Notably, 72.4% of MDR patients required antibiotic regimen adjustments, highlighting the potential value of intraoperative microbiological sampling for guiding postoperative management. These findings align with studies demonstrating that culture-based antibiotic adjustments can significantly reduce SSI rates in complicated appendicitis cases [12,18,19].

Our institution follows the Robert Koch Institute recommendations for perioperative prophylaxis, using cefuroxime and metronidazole [22]. A primary resistance to this regimen was observed in 11.4% of patients, particularly those with diabetes mellitus. While third-generation cephalosporins may be considered in select cases, our findings support the continued use of second-generation cephalosporins for most patients.

Currently, no international consensus exists on postoperative antibiotic therapy for appendicitis. While uncomplicated cases rarely require antibiotics, complicated cases often receive prolonged treatment. A multicenter randomized trial demonstrated that two days of postoperative intravenous antibiotics for complex appendicitis was as effective as five days in terms of infection rates and the ninety-day mortality [17]. In our cohort, nearly all complicated cases received at least three days of antibiotics, with an escalation required in 12.4% due to resistance patterns. Importantly, patients with MDR infections were significantly more likely to require adjustments in their antibiotic regimen compared to non-MDR patients. Our data emphasize the importance of intraoperative swabs for tailoring postoperative therapy, especially in complicated appendicitis. Based on antibiogram results, 21% of patients were successfully transitioned to targeted oral antibiotics, enabling an earlier discharge and outpatient management.

Previous studies, including those by Son et al. and Hu et al., similarly reported that approximately one-third of patients with complicated appendicitis exhibited resistance to initial empirical therapy [12,23]. Nevertheless, the overall benefit of routine intraoperative cultures remains debated, as some retrospective studies have suggested that adjustments based on culture results are infrequent and may not significantly reduce the postoperative morbidity [24,25]. A recent study investigating antimicrobial susceptibility in pediatric patients with acute appendicitis further underscored this controversy. While commonly used antibiotic regimens—amoxicillin/clavulanic acid or aminoglycoside/metronidazole—were appropriate in only 40% of cases, ceftriaxone/metronidazole and piperacillin/tazobactam achieved over 90% appropriateness. Despite this marked difference in the targeted efficacy, no significant reduction in complication rates was observed between the groups [26]. However, it is important to note that this study did not exclusively focus on complicated appendicitis. Since postoperative antibiotics are generally not required for uncomplicated cases, particularly in adults, these findings may have limited applicability to adult patients with complex appendicitis [26].

Taken together, these results from our prospective study suggest that rectal swabs have limited clinical relevance in guiding antibiotic therapy for AA, particularly in non-operative cases. Meanwhile, the high prevalence of MDR infections in complicated appendicitis underscores the importance of targeted antimicrobial stewardship. Intraoperative swabs may help refine postoperative antibiotic strategies, reducing unnecessary antibiotic use and potentially improving outcomes in selected patient populations.

Several limitations should be acknowledged. First, this was a single-center study with a relatively small sample size of 105 patients, which may have limited the generalizability of our findings. Second, the follow-up period was restricted to 30 days, preventing an assessment of long-term outcomes. Additionally, only surgically treated patients were included, excluding a significant proportion of those with acute appendicitis who underwent non-operative management. Consequently, bacterial analysis and antibiogram testing were not possible in these patients, further limiting the applicability of our results to the broader appendicitis population.

Future prospective multicenter studies with larger cohorts and extended follow-up periods are needed to validate our findings and provide more comprehensive insights.

## 4. Materials and Methods

### 4.1. Study Design

This prospective, single-center study was conducted at the University Hospital of Goethe University, Frankfurt am Main, Germany (Theodor-Stern-Kai 7, 60590, Frankfurt), from April 2022 to August 2023, with all patients completing a 30-day follow-up. The study was approved by the Institutional Ethics Committee of the Medical Faculty of the University of Frankfurt (approval number: 42-22; case number: 2022-635), adhering to the ethical principles outlined in the Declaration of Helsinki. Prior to inclusion, all patients provided written informed consent. The study was registered in the German Clinical Trials Register under www.drks.de, with the identifier DRKS00028610 [27].

### 4.2. Patients

All patients aged 18 and above who underwent appendectomy for acute appendicitis (ICD-10 code: K35) between April 2022 and July 2023 were included in this study and prospectively documented. The exclusion criteria included negative appendectomy cases and patients treated non-operatively with antibiotics for uncomplicated appendicitis.

### 4.3. Diagnosis, Surgical Procedure, and Histopathology

Patients presenting with suspected acute appendicitis underwent clinical examination, blood and urine analysis, and an ultrasound. Appendectomy was indicated by a board-certified surgeon. Appendectomy was conducted as a standard three-port laparoscopy, with the closure of the appendix stump using an endoscopic stapler device (Endo GIA™ Ultra Universal Loading Unit with Tri-Staple™ Technology from COVIDIEN, Medtronic GmbH, Meerbusch, Germany). In cases where laparoscopic appendectomy was deemed unsafe, open appendectomy was indicated. The use of intra-abdominal drainage was at the surgeon’s discretion. All surgical specimens were sent for histopathological examination. The histopathologic criteria for diagnosing acute appendicitis included the granulocytic infiltration of the lumen or all wall layers and adipose tissue, the ulceration of the mucosa or all wall layers, and the presence of intramural abscesses or necrosis. Specimens not meeting these criteria were classified as “no appendicitis”, resulting in the exclusion of three patients due to negative histopathological findings. The time of admission was recorded in the electronic medical record by the central emergency department. The time from the symptom onset to hospital admission and the time to surgery were recorded in minutes or hours.

### 4.4. Microbiological Analysis

A swab tube (Transystem sterile transport swab, suitable for aerobes and anaerobes, COPAN, Brescia, Italy) was used to collect a culture swab from the appendix after appendectomy on the operating table. Additionally, rectal swabs for MDRO screening were taken, usually prior to surgery or antibiotic treatment. Both samples were analyzed by the Institute of Medical Microbiology and Infection Control to determine the appendiceal microbial spectrum and the MDRO presence in rectal swabs from patients with AA. All laboratory tests were performed under strict quality-controlled criteria (DIN ISO 15189:2021 standards [28]; certificate number D–ML–13102–01–00).

Cultures from patient specimens, bacterial species identification, and antibiotic susceptibility testing were performed using standard clinical microbiology methods. In particular, bacteria were identified by matrix-assisted laser desorption–ionization time-of-flight analysis (VITEK MS, bioMérieux, Nürtungen, Germany). Antibiotic susceptibility testing was carried out according to the recommendations of the EUCAST. Vancomycin-resistant *Enterococci*, methicillin-resistant *S. aureus*, and carbapenemase-resistant Gram-negative bacteria were further analyzed by molecular identification techniques including polymerase chain reaction analysis and subsequent sequencing [29].

### 4.5. Follow-Up

Clinical characteristics, including individuals’ age, sex, American Society of Anesthesiologists (ASA) score [30], body mass index (BMI), and preoperative laboratory results, as well as perioperative and postoperative data, were prospectively collected. Postoperative complications were monitored during participants’ hospital stay. Further, a 30-day follow-up was conducted via telephone, mail, or email using a standardized questionnaire. The duration of antibiotic therapy, the hospital stay, and work disability were recorded in days. Complications were identified from medical records and the follow-up questionnaire and categorized according to the Clavien–Dindo classification [15]. SSIs were classified as A1, A2, and A3 according to the Centers for Disease Control and Prevention (CDC) criteria [30].

### 4.6. Definitions

AA was classified according to Gomes [31]. Complicated appendicitis was defined as appendicitis with necrosis, an inflammatory tumor, or perforation, while uncomplicated cases showed endo-appendicitis or signs of inflammation without pericolic fluid [32]. The categorization of AA was performed according to operative records and histopathological examination.

MDROs are resistant to one or more antimicrobial drug classes according to the criteria of the CDC [33]. The Infectious Diseases Society of America has coined the acronym ESKAPE for MDROs that pose a threat to public health: multi-drug-resistant *Enterobacterales* due to the production of extended-spectrum β-lactamases (ESBLs) or carbapenemases, methicillin-resistant *Staphylococcus aureus* (MRSA), carbapenemase-producing *Klebsiella pneumoniae*, multi-drug-resistant *Acinetobacter* spp., multi-drug-resistant *Pseudomonas aeruginosa*, and *Enterococci*, especially *Enterococcus faecium* with and without vancomycin resistance (VRE) [34]. MDROs have been subdivided into multi-drug-resistant Gram-negative bacteria (MDGRN) and multi-drug-resistant Gram-positive bacteria. [35] The Robert Koch Institute’s definition was used to subdivide the MDGRN into 3MRGN/4MRGN. The focus of this classification is on resistance to antibiotics used as primary bactericidal therapeutics for serious infections (acylureidopenicillins, 3rd- and 4th-generation cephalosporins, carbapenems, and fluoroquinolones); 3MRGN have resistance to 3 and 4MRGN to 4 of the 4 groups of antibiotics [35,36].

In our cohort, patients with bacterial resistance to at least two antibiotics were classified as having multi-drug resistance (MDR), while patients with no detectable microorganisms, no antibiotic resistance, or resistance to a single antibiotic were classified as having non-multi-drug resistance (non-MDR), based on a previous study by Son et al. [12]. The complete lists of all detected microorganisms and their corresponding antibiotic resistance profiles for both cohorts are provided in the Appendix A.

An escalation to reserve antibiotics was defined as a change to imipenem, meropenem, teicoplanin, vancomycin, linezolid, or fourth-generation cephalosporins, while a general antibiotic escalation included changes to piperacillin/tazobactam, levofloxacin, or third-generation cephalosporins.

### 4.7. Statistical Analysis

The data were pseudonymized and analyzed using IBM (Armonk, NJ, USA) SPSS Statistics Subscription version 29.0.1.0 (171).

The primary endpoint of this study was to evaluate the spectrum of microorganisms, their resistance profiles, and antibiotic susceptibility in appendiceal swabs and to compare the presence of MDROs between appendiceal and rectal swabs in patients with AA. Secondary endpoints included the correlation of infection with multi-drug-resistant (MDR) pathogens with the complexity of AA, the intraoperative course, postoperative complications—especially the incidence of SSIs—the length of the hospital stay, the duration of postoperative antibiotic therapy, the response to empiric antibiotic therapy, and the duration of work disability.

Patients with MDR were compared with non-MDR patients. Interval-scaled secondary endpoints and other interval-scaled parameters were presented as medians with interquartile ranges (IQRs). For the secondary endpoint “complications”, both the total number in the respective group and the frequency of each severity level according to the Clavien–Dindo classification were reported. The Chi-squared test and Fisher’s exact test were used to examine differences in categorical variables. The Mann–Whitney U test was applied for non-dependent variables. A *p* value of less than 0.05 was considered statistically significant.

## 5. Conclusions

MDR infections in AA worsen postoperative outcomes, including causing higher SSI rates and prolonged hospital stays. Intraoperative appendiceal swabs in complicated cases may help optimize postoperative antibiotic regimens, minimize unnecessary antibiotic exposure, and reduce healthcare costs.

## Figures and Tables

**Table 1 antibiotics-14-00378-t001:** Correlation of presence of multi-drug-resistant organisms (MDROs) between appendiceal and rectal swabs.

Multi-Drug-Resistant Organisms	Only Rectal	Appendiceal and Rectal	Only Appendiceal	Total
*Escherichia coli* (ESBL-R)	3	2	0	5
*Escherichia coli* (3MRGN)	1	0	0	1
*Escherichia coli* (ESBL-R and 3MRGN)	3	1	0	4
*Pseudomonas aeruginosa*	0	1	1	2
Vancomycin-resistant *Enterococci*	1	0	0	1
*Bordetella hinzii*	0	0	1	1
Total MDROs	8	4	2	14

MDROs—multi-drug-resistant organisms; ESBL-R—extended-spectrum beta-lactamase resistance; 3MRGN—Gram-negative rods that are multi-drug-resistant to three classes of antibiotics.

**Table 2 antibiotics-14-00378-t002:** Complete spectrum of microorganisms isolated in acute appendicitis.

Appendiceal Swab	Total (n = 105)
*Escherichia coli*	49 (46.7)
-ESBL-R	2 (1.9)
-ESBL-R and 3MRGN	2 (1.9)
*Bacteroides* spp.	32 (30.4)
*Enterococcus* spp.	18 (17.1)
*Streptococcus* spp.	9 (8.5)
*Klebsiella* spp.	5 (4.5)
*Pseudomonas aeruginosa*	8 (7.6)
-3MRGN	1 (1)
*Staphylococcus* spp.	5 (4.9)
*-Staphylococcus aureus*	2 (1.9)
*-Staphylococcus lugdunensis*	2 (1.9)
*-Staphylococcus epidermidis*	1 (1)
*Cutibacterium* spp.	3 (2.9)
*Citrobacter* spp.	3 (2.9)
*Clostridium innocuum*	2 (1.9)
*Bordetella hinzii*	1 (1)
*Serratia marcescens*	1 (1)
*Schaalia turicensis*	1 (1)
*Eghertella lenta*	1 (1)
*Corynebacterium propinquum*	1 (1)
*Candida* spp.	5 (4.8)

ESBL-R—extended-spectrum beta-lactamase resistance; 3MRGN—Gram-negative rods that are multi-drug-resistant to three of four classes of antibiotics; spp.—species.

**Table 3 antibiotics-14-00378-t003:** Antibiotic resistance profiles of pathogens isolated from appendiceal swabs.

Appendiceal Swab	Total (n = 105)
**Non-MDR**	**79 (72.4)**
No resistance	59 (56.2)
Resistance	46 (43.8)
1 antibiotic	17 (16.2)
Ampicillin	6 (5.7)
Ampicillin/sulbactam	4 (3.8)
Clindamycin	3 (2.9)
Fosfomycin	2 (1.9)
Ciprofloxacin	1 (1)
Metronidazole	1 (1)
**MDR**	**29 (27.6)**
Resistance to 2 antibiotics	12 (11.4)
Penicillin G and clindamycin	1 (1)
Penicillin G and cotrimoxazole	1 (1)
Penicillin and piperacillin	1 (1)
Ampicillin and fosfomycin	1 (1)
Ampicillin and imipenem	1 (1)
Ampicillin/sulbactam and cefuroxime	3 (2.9)
Ampicillin/sulbactam and cotrimoxazole	2 (1.9)
Ampicillin/sulbactam and imipeneme	1 (1)
Piperacillin and ciprofloxacin	1 (1)
Resistance to ≥3 antibiotics	11 (10.5)
Penicillin, oxacillin, and cefuroxime	1 (1)
Ampicillin/sulbactam, ciprofloxacin, and cotrimoxazole	1 (1)
Ampicillin/sulbactam, gentamicin, and cotrimoxazole	1 (1)
Ampicillin/sulbactam, ciprofloxacin, and levofloxacin	1 (1)
Ampicillin/sulbactam, clindamycin, and cotrimoxazole	1 (1)
Ampicillin/sulbactam, cefotaxime, ciprofloxacin, and cotrimoxazole	1 (1)
Ampicillin/sulbactam, cefotaxime, ciprofloxacin, and levofloxacin	1 (1)
Ampicillin/sulbactam, piperacillin/tazobactam, and cefuroxime	1 (1)
Ampicillin/sulbactam, piperacillin/tazobactam, and clindamycin	2 (1.9)
Ampicillin, piperacillin, cefotaxime, and fosfomycin	1 (1)
MDROs according to CDC	6 (5.7)
Ampicillin/sulbactam, cefotaxime, ciprofloxacin, and gentamicin	1 (1)
Ampicillin/sulbactam, cefotaxime, ciprofloxacin, and imipenem	1 (1)
Ampicillin/sulbactam, ESBL-R, and cotrimoxazole	1 (1)
Ampicillin/sulbactam, piperacillin/tazobactam, cefotaxime, and meropenem	1 (1)
Ampicillin/sulbactam, ESBL-R, cefotaxime, and imipenem	1 (1)
Piperacillin, cefepime, levofloxacin, and tobramycin	1 (1)

CDC—Centers for Disease Control and Prevention; ESBL-R—extended-spectrum beta-lactamase resistance; MDRO—multi-drug-resistant organisms; MDR—multi-drug resistance.

**Table 4 antibiotics-14-00378-t004:** Comparison of clinical and diagnostic characteristics between non-multi-drug-resistant (non-MDR) and multi-drug-resistant (MDR) patients.

Variables	Total (n = 105)	Non-MDR (n = 76)	MDR (n = 29)	*p* Value
Alvarado score				0.91
Median	8	8	8	
Range	0–10	0–10	2–10	
C-reactive protein [mg/dL]				0.82
Mean	5.9	5.7	6.3	
Range	0–39.5	0–32.9	0.1–39.5	
Leukocytes [/µL]				0.89
Mean	13.1	13.1	13.1	
Range	0.2–26.6	0.2–26.6	6–19.6	
CT/MRI scan, n (%)	63 (60)	43 (56.6)	20 (69)	
CT	59 (56.2)	40 (52.6)	19 (65.5)	0.28
MRI	4 (3.8)	3 (2.9)	1 (3.4)	
Duration from symptom onset to surgery [hours], n (%)				**0.02**
<12	7 (6.7)	6 (7.9)	1 (3.4)	
12–24	21 (20)	19 (25)	2 (6.9)	
>24	77 (73.3)	51 (67.1)	26 (89.7)	
Complexity, n (%)				0.09
Uncomplicated appendicitis	70 (66.7)	54 (71.1)	16 (55.2)	
Complicated appendicitis	35 (33.3)	22 (28.9)	13 (44.8)	
Abdominal abscess	10 (9.5)	7 (9.2)	3 (10.3)	

CT—computed tomography; MDR—multi-drug resistance; MRI—magnetic resonance imaging.

**Table 5 antibiotics-14-00378-t005:** Classification of postoperative complications according to Clavien–Dindo grades [15].

Grade	Definition
**1**	Any deviation from the normal postoperative course without the need for pharmacological treatment or intervention
**2**	Requiring pharmacological treatment
**3**	Requiring surgical, endoscopic, or radiological intervention
**3a**	Intervention not under general anesthesia
**3b**	Intervention under general anesthesia
**4**	Life-threatening complication requiring admission to intermediate or intensive care unit
**5**	Death of a patient

**Table 6 antibiotics-14-00378-t006:** Comparison of postoperative complications and antibiotic therapy between non-multi-drug-resistant (non-MDR) and multi-drug-resistant (MDR) patients.

Variables	Total (n = 105)	Non-MDR (n = 76)	MDR (n = 29)	*p* Value
Postoperative complications (Clavien–Dindo), n (%)	17 (16.2)	8 (10.5)	9 (31)	**0.02**
Grade 1	8 (7.6)	5 (6.6)	3 (10.3)	0.52
Grade 2	2 (1.9)	1 (1.3)	1 (3.4)	0.48
Grade 3a and 3b	7 (6.7)	2 (2.6)	5 (17.2)	**0.01**
Surgical site infections (SSIs), n (%)	15 (14.3)	8 (10.5)	7 (24.1)	0.12
Superficial	10 (9.5)	7 (9.2)	3 (10.3)	0.86
Deep incisional	1 (1)	0	1 (3.4)	0.28
Organ/space	4 (3.8)	1 (1.3)	3 (10.3)	**0.03**
Length of hospital stay [days], median (IQR)	3 (3)	3 (3)	4 (5)	**0.03**
Postoperative antibiotics, n (%)	50 (47.6)	31 (40.8)	19 (65.5)	**0.03**
Antibiotic duration [days], median (IQR)	7 (5)	6 (3)	8 (7)	0.46
Antibiotic switch, n (%)	23 (21.9)	12 (15.8)	11 (37.9)	**0.02**
Switch to oral antibiotics	13 (12.4)	9 (11.8)	5 (26.3)	0.31
Antibiotic escalation	13 (12.4)	5 (6.6)	8 (27.6)	**0.01**
Type of antibiotics used, n (%)				
Cefuroxime/metronidazole	23 (21.9)	16 (21.1)	7 (24.1)	0.79
Piperacillin/tazobactam or 3rd generation ceph.	7 (6.7)	3 (3.9)	4 (13.8)	0.09
Reserve antibiotics *, n (%)	6 (5.7)	2 (2.6)	4 (13.8)	**0.048**

IQR—interquartile range; ceph—cephalosporins; MDR—multi-drug resistance, SSI—surgical site infection; 3rd—third. * Imipenem, meropenem, teicoplanin, vancomycin, linezolid, or fourth-generation cephalosporins.

## Data Availability

The data presented in this study are available upon request from the corresponding author after approval by the University Hospital’s Data Protection Integration Center.

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
