# Peer review of "A Prospective Analysis of the Burden of Multi-Drug-Resistant Pathogens in Acute Appendicitis and Their Implication for Clinical Management"

_antibiotics, 2025, doi:10.3390/antibiotics14040378_

Round 1
Reviewer 1 Report
Comments and Suggestions for Authors
Dear Authors,
This research paper is well presented and academically articulated to ensure clarity and accessibility. Acute appendicitis is among the most frequent causes of abdominal pain requiring surgical intervention. Antibiotic therapy plays a crucial role in preventing infections that may occur postoperatively. However, there is no unanimous consensus in the literature regarding the optimal antibiotic regimen or its influence on the incidence of postoperative complications. Therefore, in the era of antibiotic resistance, studies exploring the spectrum of antibiotic activity provide valuable contributions to this area of research. I thus report a few comments to improve this paper making it more available for others researchers and readers.
- Besides surgical site infection and abdominal abscess, were any other postoperative complications reported? Is there a correlation between the occurrence of abdominal abscess and the MDR group?
- Did the authors find a correlation between the MDR group and patients presenting with complicated acute appendicitis?
- In light of the obtained results, do the authors consider it useful to perform a rectal swab as part of the management of these patients?
- During the study period, how many patients with uncomplicated acute appendicitis were treated conservatively with antibiotic therapy? For these patients, bacterial analysis and antibiogram testing were not possible, thereby excluding a significant portion of patients with acute appendicitis. It is recommended that these considerations be included in a section discussing the limitations of the study.
- The role of antibiotic therapy in cases of acute appendicitis remains a debated topic in the literature. Therefore, further studies are needed to analyze the antibiotic resistance of bacteria responsible for this condition and to compare the obtained results in both pediatric and adult populations. I suggest you read the paper “The Care of Appendicular Peritonitis in the Era of Antibiotic Resistance: The Role of Surgery and the Appropriate Antibiotic Choice” by Di Mitri et al. which reports the result of the antibiograms with one of the highest number of samples in literature.
- In line 140, where the Clavien-Dindo classification is mentioned, including a table summarizing this classification could facilitate the comprehension of the content.
- I suggest to put the “Materials and Methods” paragraph before the “Results” paragraph to better understand the study design and the results obtained.
- Reference numbers are correctly placed in square brackets, but they have to be placed before the punctuation, not after.
- Line 22 “scarce” is not incorrect, but in scientific language the word “limited” is preferable.
- Line 41 colonization is a singular term, please change “were associated” with “is associated”.
- Line 61 “buit” is a typo, please correct with “but”.
- Line 68 “impact” is a singular word, so please correct the verb with “is conflicting”.
- Line 269 change “was” with “were” for agreement with the plural subject.
- Check reference number 6 between 2 and 3.
- Check the format of the reference 13.
- Please check the manuscript once again in terms of English preferably by a native speaker since there are a few minor grammatical errors or typos.
I suggest to check the manuscript once again in terms of English preferably by a native speaker since there are a few minor grammatical errors or typos.
Reviewer 2 Report
Comments and Suggestions for Authors
Consider rephrasing the title for clarity and fluency: "Prospective Analysis of the Burden of Multi-Drug Resistant Pathogens in Acute Appendicitis and Their Implications for Clinical Management"
Revise sentences in abstract for better flow. Instead of "Multi-drug resistant colonization in appendicitis are associated with worse outcomes," use "Colonization with multi-drug resistant organisms in appendicitis is associated with worse outcomes."
Some sentences are lengthy and repetitive. Aim for brevity while maintaining content.
It’s good that data is presented, but it should be clearer. For example, "MDROs were identified in 11.4% of rectal swabs but showed limited predictive value"—explain briefly what the limited predictive value means.
The background is well-researched and contextualizes the study.
Streamline the rationale for the study. The introduction feels slightly redundant in some places.
“Buit” should be “but” in “buit current antimicrobial guidelines...”
The aim of the study is stated at the end, but consider making it stand out more clearly as a hypothesis or primary research question.
In methods, clarify "single university-affiliated institution" early on. The hospital name and location should be specified in the methods. The description is adequate but would benefit from the mention of quality control standards earlier.
Better define the primary and secondary endpoints upfront. Currently, the statistical section is clear but would benefit from more details on sample size calculation (if applicable) and power considerations.
The results are comprehensive and include relevant data. However, you should improve table clarity: Ensure all abbreviations are explained below each table, even if explained elsewhere. Consider reorganizing tables for better readability.
Sentences such as "They required more frequent postoperative antibiotics" should be: "These patients required more frequent postoperative antibiotic treatment."
In Discussion start by summarizing the key findings before moving into interpretations. The discussion compares the findings with existing literature but could expand on why the findings differ or align. The limitations section is missing or minimal. Explicitly discuss limitations such as sample size, single-center design, and lack of long-term follow-up. A clearer statement on future research directions would strengthen the discussion.
Comments on the Quality of English Language- The manuscript addresses an important and clinically relevant topic: the prevalence and clinical implications of multi-drug resistant organisms (MDROs) in acute appendicitis. The prospective nature of the study strengthens its impact.
- However, the manuscript could benefit from language polishing and some reorganization to improve clarity and readability.
- Some grammatical mistakes and stylistic issues are present throughout the text. A thorough proofreading or professional language editing is recommended.
Round 2
Reviewer 2 Report
Comments and Suggestions for Authors
The authors have responded satisfactorily to the demands suggested by the reviewers. In my opinion, the manuscript can be considered ready for publication in its current version.